# An Agent-Based Statistical Physics Model for Political Polarization: A Monte Carlo Study

**DOI:** 10.3390/e25070981

**Published:** 2023-06-27

**Authors:** Hung T. Diep, Miron Kaufman, Sanda Kaufman

**Affiliations:** 1Laboratoire de Physique Théorique et Modélisation, CY Cergy Paris Université, CNRS, UMR 8089 2, Avenue Adolphe Chauvin, 95302 Cergy-Pontoise, France; 2Department of Physics, Cleveland State University, Cleveland, OH 44115, USA; m.kaufman@csuohio.edu; 3Levin School of Urban Affairs, Cleveland State University, Cleveland, OH 44115, USA; s.kaufman@csuohio.edu

**Keywords:** political polarization, statistical physics model, Monte Carlo simulation, anticipatory scenarios

## Abstract

World-wide, political polarization continues unabated, undermining collective decision-making ability. In this issue, we have examined polarization dynamics using a (mean-field) model borrowed from statistical physics, assuming that each individual interacted with each of the others. We use the model to generate scenarios of polarization trends in time in the USA and explore ways to reduce it, as measured by a polarization index that we propose. Here, we extend our work using a more realistic assumption that individuals interact only with “neighbors” (short-range interactions). We use agent-based Monte Carlo simulations to generate polarization scenarios, considering again three USA political groups: Democrats, Republicans, and Independents. We find that mean-field and Monte Carlo simulation results are quite similar. The model can be applied to other political systems with similar polarization dynamics.

## 1. Introduction

Political polarization in democracies has been a subject of intensive studies in recent years [1,2,3,4,5,6]. Polarization stems from individuals forming broad and encompassing clusters organized around cohesive packages of polotical beliefs [2]. “Political polarization” in a word is the distance between the “left” and the “right” [3,4]. The case of the United States is striking: political polarization has rapidly increased in recent years [6,7,8,9,10]. European democracies were no exception [11]. In the USA, this tendency began in the 1970s [12] as reflected in polls [9]. Political polarization did not only affect the population, but also the media [13]. Each party proposed policies to solve societal problems ranging from government aid to the needy, race, immigration, national security, and the environment [6]. However, conflict over solutions to societal problems led to more problems in democracies: political polarization has serious deleterious societal [1,14,15,16] and economic [3] consequences. One of these is that people gradually lose the ability to work together, to compromise, make, and implement deals. In time, this can lead to societal breakdown [15,17].

One notable problem caused by polarization is that individuals increasingly tend to believe only in scientific information which justifies their political perspective, regardless of its factual basis [15,18,19]. Another negative consequence of polarization is that needed change can no longer occur through debate and persuasion. Instead, the party in power imposes its policies which will be undone as soon as the other party prevails [15] leading to instability and to collective and individual losses. Since polities are complex systems [20] within which interactions change with time, according to [21,22], empirical studies do not suffice to help us understand political polarization dynamics. We also need theoretical modeling to help explore conditions under which an event can happen. Agent-based modeling has great potential in this regard [23,24]. Modeling can help at least prepare information that might be useful in reducing the impacts of polarization [25,26].

Amongst many methods used to investigate polarization, sociophysics—namely, applying physics tool to the study of social phenomena—has been a very effective approach. It can handle complexity in various domains, including politics, and provide insights complementing those gleaned from other disciplines [26]. One sociophysics approach to modeling complex systems uses network models [25,27,28,29,30]. Such models have been called “generative” as opposed to inductive or deductive and are arguably well-suited to assist decision making [31]. They have already been used in studies of polarization (e.g., [8,18,32,33,34]). Despite their seeming conceptual simplicity (compared to traditional, complicated, data-intensive multi-variate regression models prevalent in social sciences) network models allow testing of hypotheses regarding such issues as the role of the media in social dynamics (e.g., [30]), or the use of bot agents to alter public opinion [35]. Network models can be used to explore polarization trends and find avenues for intervention, to avert some of its negative consequences. Since prediction in the context of a complex system is at best limited [20], generating qualitative anticipatory scenarios which can be queried (e.g., [36,37,38,39,40,41]) is an alternative. This entails producing a range of possible outcomes with respect to a variable of interest, and mapping their consequences and exploring intervention possibilities. The general network approach has gained currency in the first decades of the century, as scholars have applied it to a variety of contexts. For example, using network models we have anticipated election outcomes in the US and in Bosnia–Hercegovina [26,40], and we examined various outcomes of labor management contract negotiations in France [41]. As Ref. [42] has argued, anticipatory scenarios are useful in supporting the development of robust strategies of action in the face of high levels on uncertainty characterizing complex systems.

Within Western democracies political polarization is on the increase, undermining collective decision making abiity (e.g., [15,43,44]. We have examined polarization dynamics in the USA between Democratic- and Republican-affiliated individuals, using an agent-based model borrowed from statistical physics and mean-field theory (see Kaufman et al. [45] in this issue). We considered the effect of non-affiliated independents [9,10] who sometimes lean toward one of the two other groups in specific conflicts and during elections. Using that agent-based model [45] and past patterns [43,44], we generated scenarios of the three groups’ attitude trajectories in time. Since in [45], polarization trends continued unabated in the absence of any intervention, we explored what might reduce polarization, such as leaders bringing people together, or focusing events such as natural disasters.

In our previous article, agents’ interactions had a wide range (mean-field), meaning that each individual interacted with each of the others. Here, we extend our work by assuming, instead, that individuals interact only with their “neighbors”. We explore the insights to be gained with the short-range interactions assumption on a Bravais lattice, which may be more realistic in terms of how individuals communicate and try to persuade others of their political stance. Moreover, this kind of short-range interaction matches a “massively parallel” approach proposed by [46] as a means of reducing polarization. It consists of numerous individuals acting purposely in their physical, social, and professional neighborhoods to combat polarization by breaking through the currently prevailing acute homophily [5] and engaging with out-groups (people with differing political affiliations and perspectives than their own). Our model may help assess the extent to which the massively parallel approach can be effective in reducing polarization. Note that agent-based modeling has been used to study attitude change in societies [47].

In Section 2, we describe the initial model [45] and the Monte Carlo simulation approach we use here. Then in Section 3, we present the simulation results and discuss their meaning in terms of polarization trends scenarios. We conclude in Section 4 with a summary of findings and some directions for future developments.

## 2. Model and Method

### 2.1. Model

In our recent paper in this issue ([45]), we used agent-based modeling to extend a sociophysics two-group network model of conflict dynamics [36] to three political groups in the US: Democrats (group 1), Republicans (group 2), and Independents (group 3). Groups 1 and 2, homophilic [5,6] drive polarization, which we have measured as the gap between their average attitudes toward a composite of key political issues (such as how the economy and the environment should be managed). Although the Independents (group 3) are unaffiliated, they matter: since 2004 their share of the adult population has ranged between 27% and 50% (most recently, in February 2023, 44%, [43]), and at any moment in time they may lean toward, and strengthen group 1 or group 2 [43,44]. Thus, they represent recruitment pools for the other two groups, and gain importance especially at election times.

To describe the political stance of an individual in a society, we use a spin *S* model, where the attitude *S* ranges between −1 (extreme left) and 1 (extreme right). An individual’s stance can take any value on the continuum within this range. Such a model is called a “continuous Ising model” ([48,49]), as opposed to the discrete Ising model, where *S* could only take two values, −1 and 1. We used this continuous *S* model to examine polarization trends within the framework of the mean-field theory (MFT) (Kaufman et al. [45]). Here we use Monte Carlo (MC) simulations with short-range interactions between individuals, with real-time fluctuations, and then we compare the MC results with those of MFT, where long-range interactions with no fluctuations are the basis of the approximation.

Each individual in group *i* (i=1,2,3) has a stance compatible with the group’s attitude $S_{i}$ regarding a specific issue under debate—economics, social issues, defense, etc.—or (here) a package of such issues (in the [1,6] sense). The individual stances have values between −1 and +1, where −1 corresponds to the democrats/progressive/left position (i=1), while +1 corresponds to the republicans/conservative/right position (i=2). Individuals thus align with the group whose average stance is compatible and closest to their own [1].

Inside groups 1 and 2, individuals are homophilic [5]; they tend to prefer to communicate with each other, rather than with individuals from a different group. We denote $J_{i}$ the link between members of group *i*. It quantifies the cohesiveness of group *i*. Through $J_{i}$, members inside each group attempt to persuade each other of their own stance, effectively diminishing intra-group differences and causing stances to converge.

Individuals in each group also keep an eye on the other groups’ average attitudes, which in turn influence their own, either nudging the group average to a more extreme value or to a more moderate one. These inter-group interactions are described by parameters $K_{ij}$. For group 1, the inter-group interaction terms, −$K_{12}S_{1}<S_{2}>$ and −$K_{13}S_{1}<S_{3}>$, represent the influence of the mean stances of groups 2 and 3, $<S_{2}>$ and $<S_{3}>$ respectively, on an individual in group 1. The inter-group interactions $K_{12}$ and $K_{21}$ are not necessarily equal. At times, members of one group may feel cooperative toward the other, who might not reciprocate. Therefore, in general, $K_{ij}\neq K_{ji}$. While physics phenomena obey Newton’s third law, the magnitudes of human action and reaction do not have to be equal. Rather, the effect of group *i* on group *j* can be different in magnitude and sign from the effect group *j* has on group *i*. Hence our model is not described by a single Hamiltonian and its dynamics is not the Glauber dynamics (our spin is not Ising +/−1 but is continuous). A temperature, reflecting contextual factors, drives the variability in individual preferences in a group. Our dynamic model captures the evolution of group preferences by assuming that the intensity of interactions involves the product of individuals’ preferences at a current time and average preferences of opposing groups at an earlier time. This lag reasonably reflects the fact that results of individual persuasion efforts in one time-period materialize at a later time.

The intra-group cohesion parameters *J* and the inter-group influence parameters *K* affect the average group attitudes in time. For instance, according to the recent Gallup polls [43], in early 2023, 40% of adults declared themselves independent—with zero internal cohesion $J_{3}$, since they are not organized or formally linked, like Democrats or Republicans, but rather a bin for the non-affiliated. However, in February 2023 all but 7% of them leaned either Democrats or Republicans, at least partly in response to persuasion efforts by the other two groups.

We also use a magnetic field $h_{i}$ to represents the effect of group i’s leadership on group’s members. When $h_{i}>0$, group i’s mean stance is nudged toward positive values; when $h_{i}<0$ the mean stance is nudged to negative values.

In [45] we defined a polarization measure *P* as the distance between the mean stances of groups 1 and 2 at a given time *t*:(1)$$P=(<S_{2}>-<S_{1}>)/2$$
such that −1≤P≤1. $<S_{i}>$ is the average individual stance of group *i* calculated at a time *t*. When P=0, there is no polarization. It occurs when groups 1 and 2 have equal average stances $<S_{1}>=<S_{2}>$. When P=1, polarization is extreme (also called hyperpolarization (e.g., Burgess et al. [15]). This can occur when Democrats’ stance $<S_{1}>=-1$ (most progressive/left) and Republicans’ stance $<S_{2}>=1$ (most conservative/right). It can also occur, rather paradoxically, when P=−1, because Democrats’ stance $<S_{1}>=1$ and Republicans’ stance $<S_{2}>=-1$.

The general model has twelve parameters: three *J*, one for each of the three groups’ respective internal cohesiveness, six *K* capturing the three groups’ relationships with each other, and three $h_{i}$ to describe leadership effects, if any. The *J* and *K* parameters can be selected qualitatively, as we have done here, using publicly available poll data (see [6,31,37,45]).

We solved this model using the mean-field approximation [45]. One finding was that Independents (group 3) can alter the result of an election. Here, we use a similar model, but with short-range intra-group interactions and perform MC simulations. The model’s Hamiltonian of group *i* is:(2)$$H_{i}\left(t\right)=-J_{i}\sum_{m,n}S_{i}\left(m,t\right)\cdot S_{i}\left(n,t\right)-h_{i}\sum_{m}S_{i}\left(m,t\right)$$
where *i* indexes group *i*, and $S_{i}\left(m,t\right)$ is the stance of an individual *m* in group *i* at time *t*. The sum is performed over the nearest neighbors (NN) *m* and *n* belonging to group *i*. Note that for group 3 (Independents), the first sum is zero because $J_{3}=0$. Also, a group at *t* interacts with the average stances of the other groups at t−1.

When the three groups interact, the Hamiltonians of each group is as follows: (3)$$\begin{array}{ccc} H_{1}\left(t\right) & = & H_{1}\left(t\right)-K_{12}\sum_{m}S_{1}\left(m,t\right)<S_{2}\left(t-1\right)>-K_{13}\sum_{m}S_{1}\left(m,t\right)<S_{3}\left(t-1\right)> \end{array}$$(4)$$\begin{array}{ccc} H_{2}\left(t\right) & = & H_{2}\left(t\right)-K_{21}\sum_{m}S_{2}\left(m,t\right)<S_{1}\left(t-1\right)>-K_{23}\sum_{m}S_{2}\left(m,t\right)<S_{3}\left(t-1\right)> \end{array}$$(5)$$\begin{array}{ccc} H_{3}\left(t\right) & = & H_{3}\left(t\right)-K_{31}\sum_{m}S_{3}\left(m,t\right)<S_{1}\left(t-1\right)>-K_{32}\sum_{m}S_{3}\left(m,t\right)<S_{2}\left(t-1\right)> \end{array}$$

Before proceeding to the simulation method, let us discuss the role of the “political” temperature *T* which we introduce below, and which we borrowed from statistical physics. There, the temperature represents thermal agitations of the particles (spins, for example). Thus, T acts as a disordering factor: at low *T*, particles stay in the lowest energy state (or very close to it), while at high *T*, they vigorously change their state in an independent manner, causing disorder in the system in spite of the inter-particle interaction which favors order. Examples in physics are numerous. Here are a few: coupled atoms in a crystal are ordered at low *T* and they cause melting at high *T*; spins in ferromagnets are parallel at low *T* but become disordered at high *T*. In the context of political groups considered here, T represents the political ambiance of the society. When an election is not imminent, or in otherwise calm, prosperous times, *T* is low. Each group is relatively stable, with no significant effect of inter-party interaction. Close to an election or during politically fraught times with important issues at stake (such as strained economies or international tensions), intra-party cohesiveness may wane, due to the fluctuation of individual stances of its members, equivalent to high “political” temperature *T*. Then, each group might attempt to take advantage of the weakened cohesiveness of the other groups to enhance its influence in the competition. As we shall see below, *T* plays an important role in outcomes of political contests.

### 2.2. Simulation Method

Of the three groups, 1 (Democrats), 2 (Republicans), and 3 (Independents) in the US political system, whose interactions we model, we assume group 1 to have a stronger cohesiveness (largest *J*), and to be governing. Group 2 has weaker cohesiveness and is in opposition. Group 3 is composed of individuals having no unified political framework or formal communication links. The independents are at times (though not always) attracted to the stances of the opposition party, playing a contrarian role.

For the MC simulations, we represent each of the three groups with a triangular lattice of size N×N, where each site is occupied by a member. Each member interacts with its six nearest neighbors (NN) at time *t*, and considers the average stances of the other groups calculated at *t* – 1 (a realistic lag). Regarding the choice of lattice, here we have applied the Monte Carlo methodology testing it on a relatively simple geometry: the triangular lattice. The choice of this lattice allows for a maximum number of NN in 2D. We can use a 3D lattice to have more NN such as a FCC or a HCP with 12 NN. However we believe it will not give new phenomena. For each group, we use the periodic boundary conditions to reduce the size effects. In general, we take the size of 60 × 60 lattice sites for each group. See Figure 1 for the interaction parameters described in the previous section. Note that the notion of NN interactions in politics does not necessarily mean that people are geometrically close to each other. Rather, it refers to the number of people generally in contact with an individual.

To simulate the three groups’ interactions, we first thermalize each group to determine their intrinsic cohesiveness at a given *T*. The simulation is done as follows: for a group we generate initial individual stances −1 (Democrats) for group 1, 1 for group 2 (Replublicans), and 0 (Independents) for group 3. We then use the Metropolis algorithm to find their collective state separately (no inter-group interactions) at a given *T* using the Hamiltonians Equation 2. When they are at equilbrium, we turn on the inter-group interactions $K_{ij}$ and we follow the evolution of each group with time *t* taking into account its interactions with the average stances of two other groups calculated at time t−1 (MC time).

The Metropolis algorithm used for updating the individual stance of a member is as follows: at a time *t*, we calculate the interaction energy $E_{old}$ of a member with its NN and with an effective field resulting from the two other groups at time t−1. We make a trial change of its state by choosing a random stance between −1 and 1. We calculate the member’s trial new energy $E_{new}$. If $E_{new}<E_{old}$, the trial state is accepted. If $E_{new}>E_{old}$, it is accepted with the probability $\exp[-\left(E_{new}-E_{old}\right)/\left(k_{B}T\right)]$. We repeat this updating procedure for all individuals in each of the three groups. Note that the Metropolis algorithm obeys the detailed balance only when the system is at equilibrium, namely when there is a probability conservation: state A to state B has the same probability with that from B to A. Our purpose is to study the time dependence of the polarization, so there is no such probability conservation. Note that there are several popular dynamics such as Glauber dynamics and Kawasaki dynamics, but to our knowledge all of them have been devised for discrete spins, not for continuous spins used in this paper. The advantage of the Metropolis algorithm is that it does not depend on the nature of spin, it can be used for any kind of spin such as continuous spins used here, XY spins or Heisenberg spins.

## 3. Results and Discussion

As seen, our model has nine principal interaction parameters $J_{i}\left(i=1,2,3\right)$ and $K_{ij}$ (i≠j,i=1,2,3,j=1,2,3) in addition to $h_{i}\left(i=1,2,3\right)$. However, in applications the choice of the parameters is limited. As discussed in [45], this choice is guided by polls [43,44] and by political common attitudes of the people: to produce anticipatory scenarios of polarization, we made the following assumptions:-The Democrats (group 1) are more cohesive than Republicans (group 2), i.e., $J_{1}>J_{2}$;-Independents (group 3) have no cohesion ($J_{3}=0$) because they have no structure or means of identifying with each other, do not communicate, and do not recruit; therefore, they exert no influence on the other two groups and, as such, $K_{13}=K_{23}=0$;-Independents tend to be contrarian to the party in power (here, group 1), thus $K_{31}<0$, and are not influenced by the opposition party, thus $K_{32}=0$ (see [50,51,52] of other examples of contrarian used in a model).

With respect to parameter value selection, guided by media and professional, longitudinal polling reports, we assigned parameter values such that they qualitatively mimic general polls results [43,44]. To enable a comparison of MC results with those obtained with the MFT model [45], we selected the same values for parameters *J* and *K*, as follows:-Intra-group interactions: $J_{1}=5$, $J_{2}=3$, $J_{3}=0$-Inter-group interactions: $K_{12}=-4$, $K_{21}=-5$, $K_{13}=0$, $K_{31}=-3$, $K_{23}=0$ and $K_{32}=0$.

Note that a negative $K_{ij}$ indicates hostility (or resistance) of group *i* toward group *j*, while a positive sign indicates attraction or potential agreement between two groups. Note that a variation of the above values respecting their signs will not alter qualitatively the results shown in the following.

Each group is thermalized at temperature *T* in order to determine the temperature range in which each group has a cohesiveness. The initial configuration is that of the lowest energy (namely at T=0, all Democrats are −1 and all Republicans +1). The following quantities have been calculated:-Cohesive energy per individual $E_{i}\left(T\right)=<H_{i}\left(T\right)/N^{2}$ where $<H_{i}\left(T\right)>$ is the thermal average at *T* given by
(6)$$<H_{i}\left(T\right)>=\sum_{t=t1}^{t2}H_{i}\left(t\right)/\left(t2-t1\right)$$
where t1 is the starting averaging time and t2 the averaging end time, taken after the equilibrating time $≃10^{5}$ MC step/spin with $t2-t1=10^{5}$ MC steps/spin,-Stance of each group (sublattice magnetization) as a function of *T*:
(7)$$M_{i}\left(T\right)=<S_{i}\left(T\right)>=\sum_{t=t1}^{t2}\sum_{n}S_{i}\left(n,t\right)/\left(t2-t1\right)/N^{2}$$
where *n* belongs to group *i*. Within the assumption of the parameters given above one has $M_{1}<0$, $M_{2}>0$ and $M_{3}>0$. We define the strength of group *i* by $Q_{i}\left(T\right)=$$|<S_{i}>|,$-Susceptibility or fluctuations of the stance of group *i* at *T*:
(8)$$\chi \left(T\right)=\left[<M_{i}{\left(T\right)}^{2}>-<M_{i}\left(T\right){>}^{2}\right]/\left(k_{B}T\right)$$

The equilibrium cohesive energies of three groups are shown in Figure 2 as a function of temperature *T*: the larger the *J* (stronger cohesiveness), the lower the energy. We see that for each group there exists a temperature at which that group becomes disordered: $T_{1}≃8.6$, $T_{2}≃8.5$, $T_{3}≃8.4$. Note that the energy of group 3 is due to its interaction with group 1. This is confirmed in Figure 3, showing the absolute values of stances $Q_{1}=\left|<S_{1}>\right|$, $Q_{2}=\left|<S_{2}>\right|$ and $Q_{3}=\left|<S_{3}>\right|$: we see that *Q* is higher for larger *J*, and drops to zero (i.e., no cohesiveness) at $T_{1}$, $T_{2}$ and $T_{3}$, respectively. In statistical physics, these temperatures are called transition temperatures [49] above which the systems become disordered. Around these temperatures, the stances of the groups strongly fluctuate as shown in Figure 4. These fluctuations of the order parameter in statistical physics correspond to the so-called susceptibilities which are the fluctuations of $M_{i}$, namely, $\left(<M_{i}^{2}>-<M_{i}{>}^{2}\right)/\left(k_{B}T\right)$.

Now, we study the dynamics of the inter-group interactions at various *T* below $T_{1}$, $T_{2}$, $T_{3}$ starting from a random initial configuration for each group. For example, let us take T=5.9746, one of the simulated temperatures. Figure 5 shows the time dependence of the groups’ stances. The stances evolve with time *t* to their stable values: $S_{1}\to -0.78$, $S_{2}\to 0.60$ and $S_{3}\to 0.15$. These are the values of the curves in Figure 3 at T=5.9746. This is an indication that our model is very robust, since convergence to the stable values needs just 200 MC steps/individual. Note that, due to the (assumed) resistance to the governing Democrats, the Independents tend to lean toward the Republicans’ stance (positive value). This in turn enhances the stance of the Republicans, giving rise to the polarization *P* which is shown in Figure 6. These results are in agreement with the results of the mean-field model [45].

Let us consider now the case where $K_{12}$ is positive, namely, Democrats can attract a number of Republicans to their stance. We use the same parameter value $K_{12}=+4$ as in Ref. [45] for comparison. The MC result shows that $<S_{1}>$ and $<S_{2}>$ oscillate with time as shown in Figure 7 at a given *T*. This is again in agreement with the mean-field theory [45].

The polarization *P*, defined in Equation (Equation 1), is shown in Figure 8, which displays oscillation: as the time evolves, the polarization changes its sign and oscillates in a regular manner. This curve is calculated at T=4.2034 but this phenomenon is seen in a large temperature region far below the transition temperatures ≃8.4. Near the transition temperatures, the oscillations are very fast and less regular, as seen in Figure 9, due to strong fluctuations of $S_{i}$ near the transition, as mentioned earlier.

The polarization observed in Figure 6 is well defined by the adhesion of the Independents to the Republicans and against the governing Democrats, giving the Republicans an advantage in elections since $P=(<S_{2}>-<S_{1}>)/2$ is very positive. In contrast, in the case $K_{12}>0$, the attraction of the Republicans to the Democrats causes an oscillation of *P* in time (see Figure 8): if an election occurs while *P* is positive, the Republicans are likely to win, while if *P* is negative at election time, the Democrats are likely to stay in power.

To conclude this section, we have performed MC simulations on the same statistical physics model as the one where we used the mean-field approximation (see [45] in this issue). Despite the fact that the mean-field model neglects fluctuations while MC simulations take into account space and time fluctuations, the two methods yield qualitatively the same patterns of political polarization.

## 4. Conclusions

We have studied political polarization between Democrats and Republicans in the USA as a function of time, using MC simulations. The model is borrowed from statistical physics where each individual is represented by a continuous Ising spin taking its values from −1 (left wing) to +1 (right wing). We have considered three groups with initial different political stances: Democrats, Republicans, and Independents. An individual within any of these groups interacts with a limited number of people sharing the same political viewpoint. At any time, individuals also consider the average stance of other groups in the previous time period, causing them to either become firmer or soften their stance. Although the model represents the political structure in the USA, it can be adapted to other three-group dynamics.

The MC simulation results show that polarization depends on the nature of the inter-group interactions. It may advantage the party in opposition and help it win an election. It may also give rise to an oscillation of the polarization (whose sign changes in time). Therefore, the outcome of an election depends on the moment in time when it occurs. One example is the Brexit referendum of 2016, modeled in [37].

It is interesting and perhaps surprising to note that the MC and mean-field models yield qualitatively very similar results, though each may also offer some additional insights into polarization dynamics. Both approaches can be used to generate scenarios that include various interventions to reduce polarization. Using the mean-field model [45], we explored effects on polarization of group leaders, and of focusing events such as severe natural disasters. The MC near-neighbor approach lends itself to generating scenarios for another kind of intervention proposed by [15,46], where it is called “massively parallel”. It consists of independent individuals and groups taking initiatives locally to reach out and initiate dialogues with people with opposite stances, thereby reducing the current acute homophily. Such initiatives are already taking place around the USA (see [46]).

We intend to explore whether (in our model) the massively parallel approach results in a longer-lasting effect than leadership, whose impact appeared rather limited in time [45]. We intend to apply the model to a non-Bravais network such as a random network with a variable number of NN that matches reality better than a triangular lattice. Note that our present model conserves each party population. It would be interesting to define the conditions under which a member of a given party can leave his party. Though rare, this corresponds to a reality. This will be considered in a near future.

In general, models borrowed from statistical physics contain sufficient ingredients to describe some complex situations in social sciences which may appear intractable (for example, in terms of number of variables and data) when studied with traditional, non-dynamic methods. New domains such as sociophysics and econophysics can help shed light on problems.

## Figures and Tables

**Figure 1 entropy-25-00981-f001:**
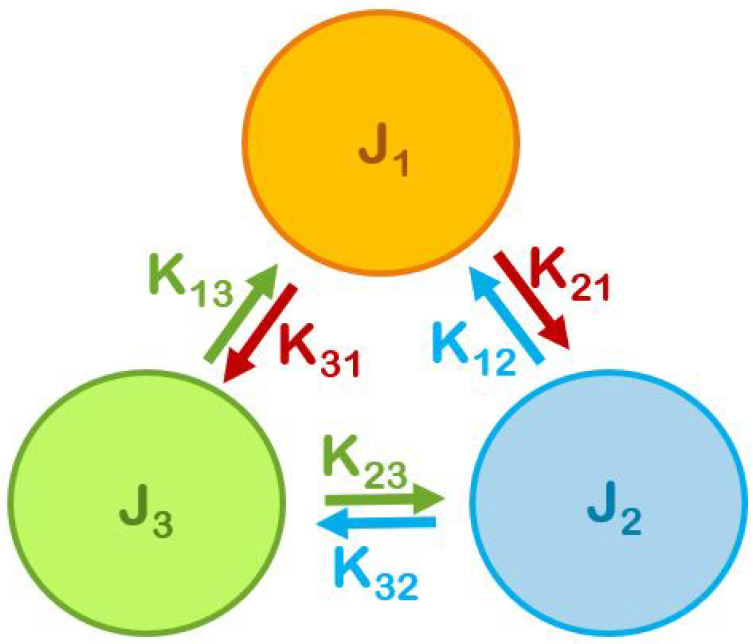
Interaction parameters. Note that $J_{3}$, intra-group interaction among the Independents, is zero. See text for comments.

**Figure 2 entropy-25-00981-f002:**
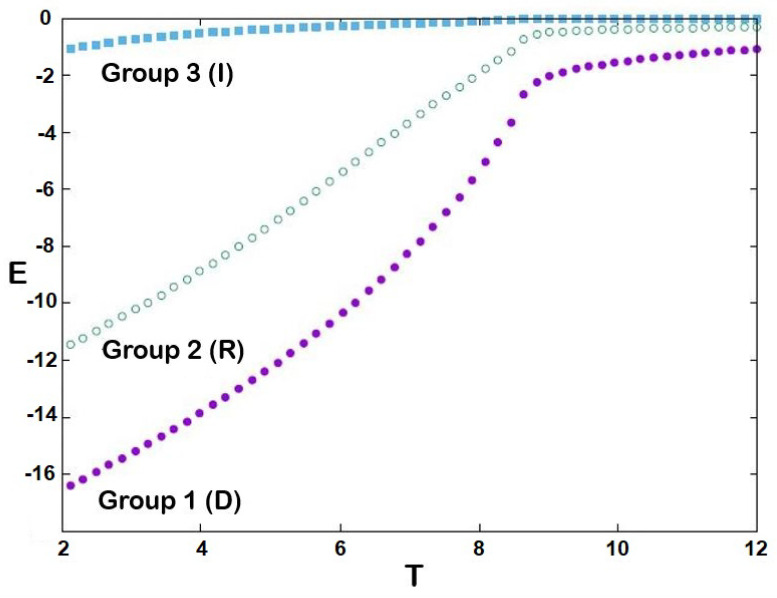
Internal energy of three groups as functions of political temperature *T*. See text for comments.

**Figure 3 entropy-25-00981-f003:**
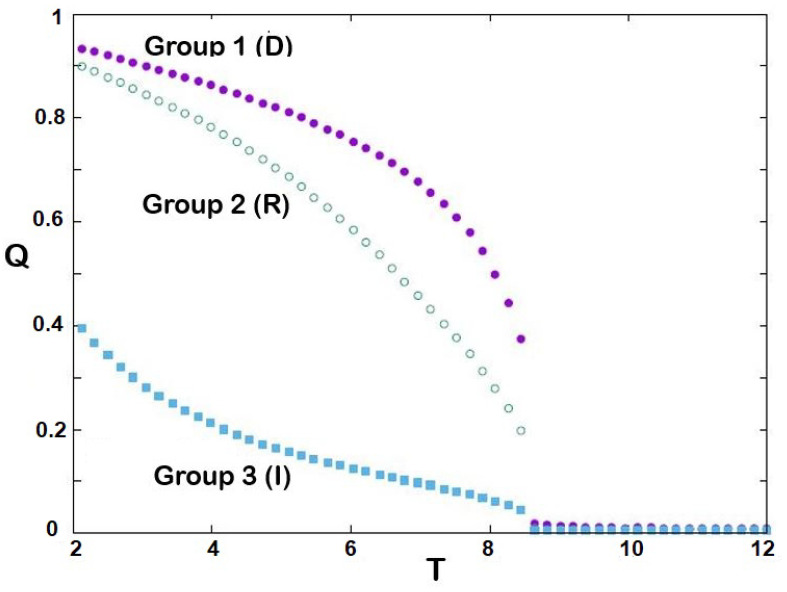
Strengths *Q* of three groups as functions of political temperature *T*.

**Figure 4 entropy-25-00981-f004:**
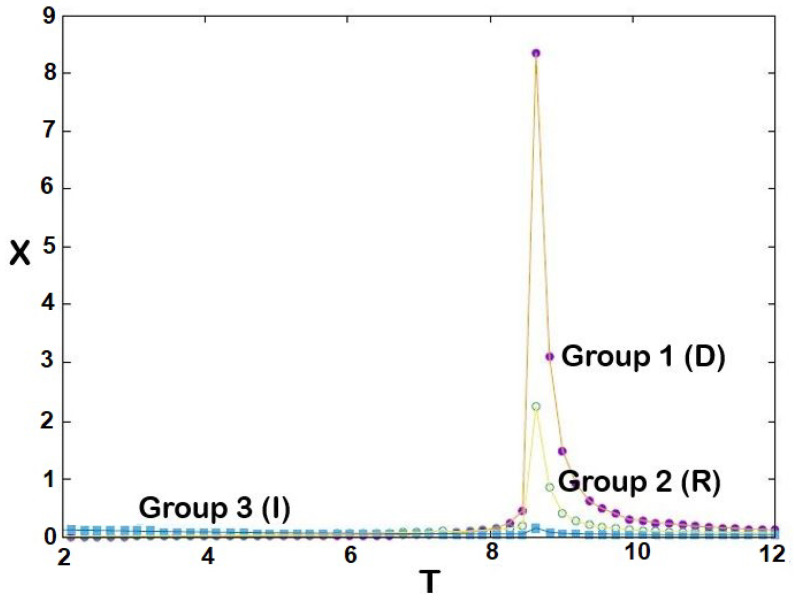
Fluctuations of the three groups’ stances as functions of political temperature *T*.

**Figure 5 entropy-25-00981-f005:**
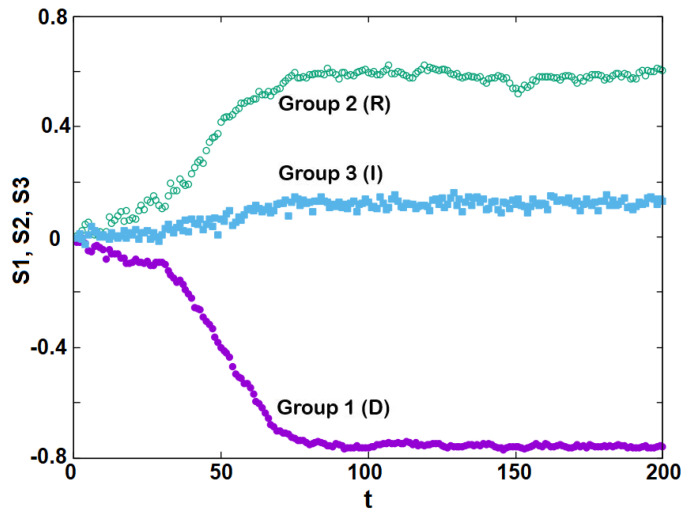
Stances of three groups as functions of time *t* taken at T=5.9746. Violet, green and blue colors correspond respectively to group 1 (Democrats), 2 (Republicans) and 3 (Independents).

**Figure 6 entropy-25-00981-f006:**
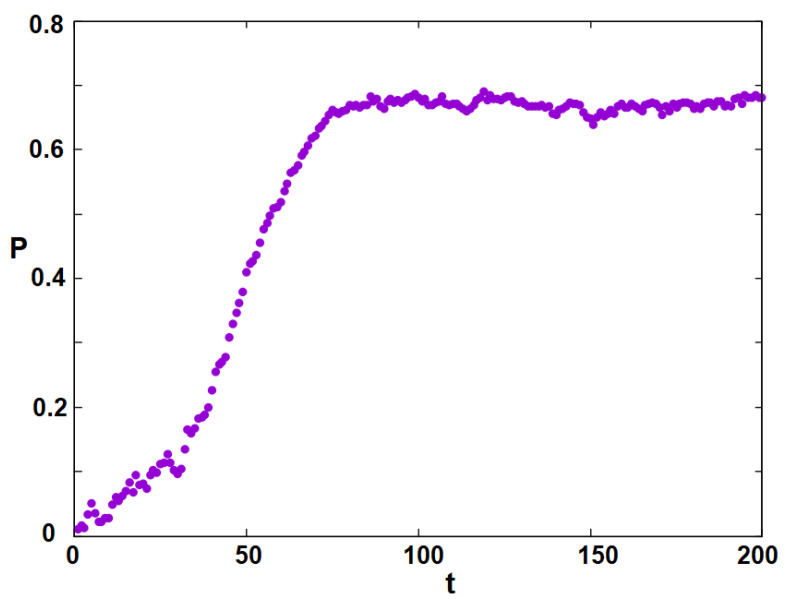
Political polarization as a function of time *t* at T=5.9746. See text for comments.

**Figure 7 entropy-25-00981-f007:**
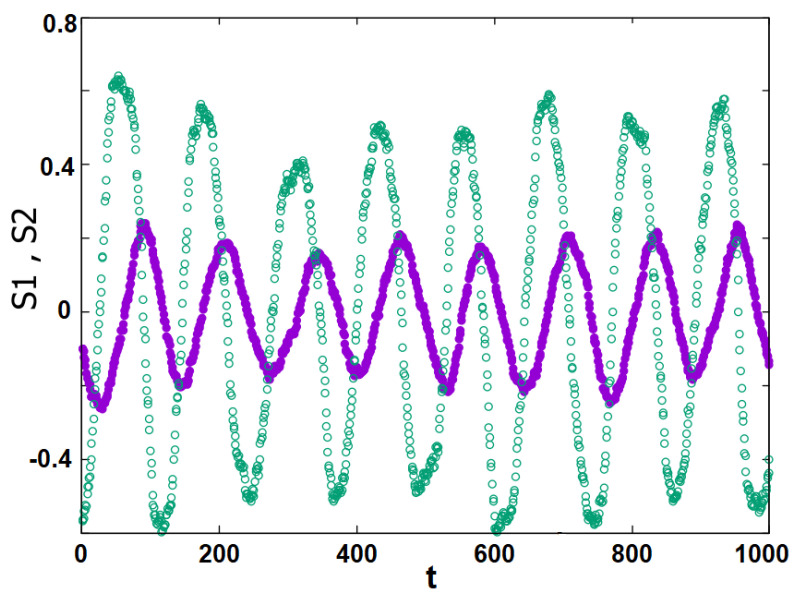
Stances $S_{1}$ and $S_{2}$ as function of time *t*, T=4.2034. The same color code as in the previous figures is used: violet for Group 1, green for Group 2.

**Figure 8 entropy-25-00981-f008:**
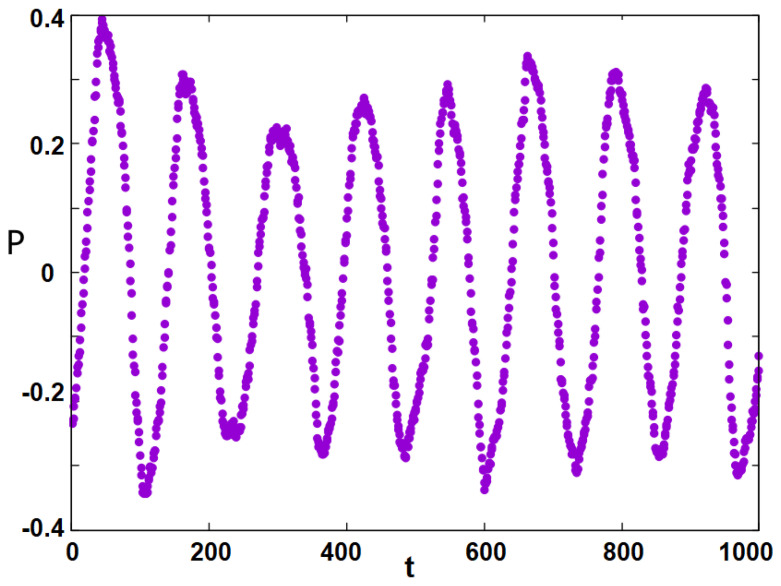
Oscillation of the political polarization *P* as function of time *t*, observed at T=4.2034. See text for comments.

**Figure 9 entropy-25-00981-f009:**
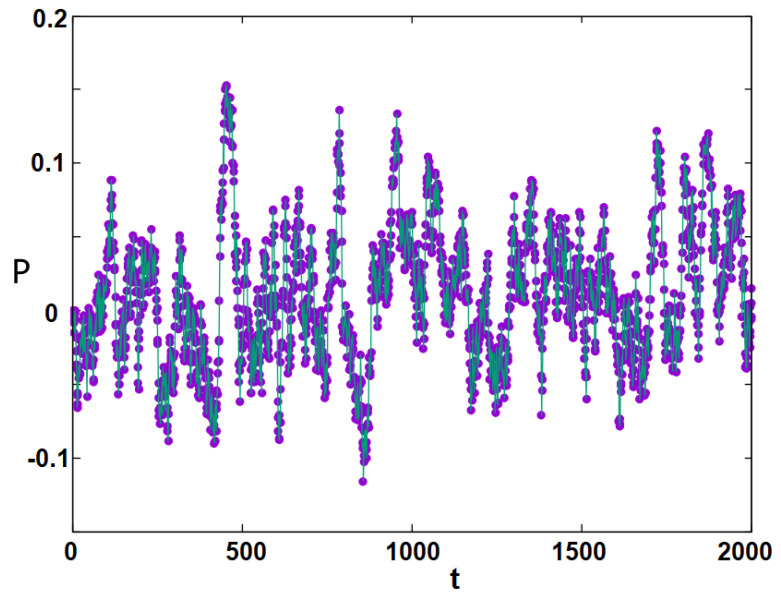
Oscillation of the political polarization *P* (circles) as function of time *t*, observed at T=6.1525. Line is guide to the eye. See text for comments.

## Data Availability

Data published in this paper are available upon request.

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
