# Peer review of "An Agent-Based Statistical Physics Model for Political Polarization: A Monte Carlo Study"

_entropy, 2023, doi:10.3390/e25070981_

Round 1

Reviewer 1 Report

A fine study.  Perhaps not overly surprising, but nice in its suggestion of local action as a possible counter-balance to polarization.

Given the use of multiple positions and local action, I was surprised that there was no reference to Axelrod's classic work on cultural polarization.  

Author Response

REPLY TO THE REVIEWERS

We thank the reviewers for carefully reading our manuscript and appreciate their comments.

Hereafter, we reply to the remarks of reviewers 1, 2 and 3 (in italic characters).  Since we added a new reference requested by reviewer 1 (new ref. 16). All the references after 16 are shifted by 1.  We have modified the text taking into account the remarks of the reviewers (in red characters). 

Reviewer 1 :

A fine study.  Perhaps not overly surprising, but nice in its suggestion of local action as a possible counter-balance to polarization.

Given the use of multiple positions and local action, I was surprised that there was no reference to Axelrod's classic work on cultural polarization.  

Reply :

Concerning reference to Axelrod:

Axelrod, R.; Daymude, J.J.; Forrest, S. Preventing extreme polarization of political attitudes. Proc. Natl. Acad. Sci. USA 2021, 118,

e2102139118.

We cited him in the previous paper (Ref. 45) in connection with the serious societal consequences of polarization.

We can cite him again in the following paragraph (the first of the introduction):

“However, instead of solving societal problems, these policies cause more problems in democracies: political polarization has serious societal [1,14,15, Axelrod(ref.16)] and economic [3] consequences.”

Reviewer 2 :

Methods and models of statistical physics proved to be in general useful for studying socio-economic phenomena. The present work intends to apply one of the basic models, the Ising model, for understanding social polarization. The two party polarization is more and more common in the political life of Western societies. The model elaborated previously by the authors, and applied also here intends to describe a system where there are two polarizing extremes and a group of independent voters, who’s political preference might be capable to decide a hypothetical election. Individuals are characterized by a continuous state variable (S) that describes their political preferences in the two-party system. The state-variables are considered as continuous classical spins that interact according to an Ising Hamiltonian, and they are also experiencing an external driving implemented in the model as an external magnetic field. The elements and their S variable are grouped in three categories: the  two polarized groups (1) and (2) and the independent group (3). There is a local interaction between
the elements in a group arranged on a triangular lattice, and a mean-field like interaction with the other groups. The Hamiltonian of the system is studied in a heat-bath, considering a Metropolis Monte Carlo method. The problem and approach is interesting, however I found many unclear issues, detailed below. Unfortunately In view of these, I cannot recommend publication of this work. My main concerns are:

  1. The system that is considered here has 12 free parameters (H_i,J_i, K_ij) + Temperature. The system is studied for one given parameter set, and I found no evidence for a general discussion on the influence of the model parameters, except the Temperature. Fixing the parameters are extremely
    subjective, and it shows an unexplained asymmetry regarding the interaction with the Independents or the two polarized groups: K_13=0, K_31=-3, K_23=0 and K_31=0. One should explain this asymmetry and also discuss the influence of the model parameters in detail.

Reply:

The matrix of inter-group interactions is not necessarily symmetrical: Kmn = Knm because of human agency. While physics phenomena obey Newton’s third law, the magnitudes of human action and reaction do not have to be equal. Rather, the effect of group n on group m can be different in magnitude and sign from the effect group m has on group n.  Hence our model is not described by a single Hamiltonian and its dynamics is not the Glauber dynamics.

A temperature, reflecting contextual factors, drives the variability in individual preferences in a group. Our dynamic model captures the evolution of group preferences by assuming that the intensity of interactions involves the product of individuals’ preferences at a current time and average preferences of opposing groups at an earlier time. This lag reasonably reflects the fact that results of individuals persuasion efforts in one time period materialize at a later time.

This has been fully discussed in our previous publications (Physica A, Refs. 36&38) where the model has been introduced. References to other sociophysics models were given there. However, to be self-contained we insert the above remark into section II. A in red characters.

  1. The Metropolis Monte Carlo method is designed and used to study equilibrium phenomena where detailed balance is preserved. It generates trustable average values for the relevant physical quantities in equilibrium conditions. The phenomenon studied here is clearly a non-equilibrium problem, where Metropolis dynamics is not justified. One could use the kinetic MC method or the Glauber dynamics, but should not use the Metropolis dynamics.

Reply: 

Note that the Metropolis algorithm obeys the detailed balance only when the system is at equilibrium, namely when there is a probability conservation: state A to state B has the same probability with that from B to A. Our purpose is to study the time dependence of the polarization, so there is no such probability conservation. Our system when evolving with time does not follow the Glauber dynamics which includes the probability conservation in the updating probability. Of course, for a system at equilibrium, the two algorithms give the same result.
We insert the above paragraph at the end of Section II.  Note that all MC simulations to study equilibrium properties of a system start with non-equilibrium configurations. The Metropolis algorithm allows the system converge to the equilibrium state. No other algorithm is equally efficient for equilibrating the system. Equilibrium state is recognized by  time-independent “averaged” physical quantities such as the internal energy. Only from the point when the system is at equilibrium, we can calculate static properties of the system, not during the equilibrating time. The Metropolis algorithm thus has been used during the non-equilibrium time to equilibrate the system in most of MC works since 1953.  Note that Metropolis algorithm has been used to study the dynamics of various systems not at equilibrium such as spin transport, relaxation-time laws in spin glasses, in frustrated  systems,…The Glauber dynamics is different from the Metropolis algorithm in  the fact that it  includes the probability conservation even in the non equilibrium state. But both satisfy the detailed balance and give almost the same result  at equilibrium except near T_c where both are bad: near T_c (|1-T/T_c|<<0.001), there are many sophisticated methods to get rid of the critical slowing down: Wolff cluster update (1989), Swendsen-Ferrenberg multihistogram techniques (1992), Wang-Landau method (2003).  However, the criticality is not the purpose of the present paper.  We hope the reviewer is convinced by this short response.

  1. One of the most problematic issue in this approach is that the elements (and their state variables) are assigned in groups from the beginning and this CAN NOT CHANGE (at least as I understood).
    For example an element from group (3) cannot leave the group and be reassigned to group (1) or (2). The same for the elements in group (1) or group (2). Their state variable will/can change, but not the group to which they belong. This means that for instance in group with initially S >0 polarization if an element would change its state variable to S<0, it would still remain in this group and interact with the
    fixed parameter set according to the Ising Hamiltonian. This is highly unrealistic.

Reply:

In our model each group is homophilic.  It is believed that individuals do not wander from one group to another often. Diffusion could be included in future studies. This point has been discussed after the citation of the Ref. 46 in the Introduction about what is called “massively parallel approach”.  The fact that an element changes its sign in a sea of elements of opposite sign is not unrealistic: some members of a party can have agreement with the opposite party at some time.

  1. I did not (or I missed it) how initially the elements are arranged on the triangular
    lattice. Elements in the same group are clustered, or independently of their group are randomly positioned? This issue should be clarified. In reality of course one might expect some spatial clustering I for the interaction network based on their political preferences.

Reply:

Each group is  thermalized at temperature T in order to determine the temperature range in which each group has a cohesiveness.  The initial configuration is that of the lowest energy (namely at T=0, Democrats -1 for everyone and Republicans +1).  This step aims at determining the temperature range in which the groups’ cohesiveness is not zero. The equilibrium states of three groups are shown in Figs. 2-4.  When the interactions are turned on, we use the random spin configuration for each group. This phrase is now added (in red characters)  in section III.

Of course, the theory of phase transition shows that there are clusters of excited people when T gets close to the critical temperature on both sides: when one decreases T from the disordered phase, small independent clusters are formed, their sizes increase and percolate at T_c, making the transition. Below T_c these clusters become larger and larger and collapse into a single one to form the perfect ordered phase at T=0. However, our purpose here is neither to study the transition behavior nor the spin landscape, but rather we study the dynamics of each group as a whole when the inter-group interactions are turned on (Figs. 5-9).

  1. In line 159 the authors state” We solved the model....”. It is unclear for me what it is meant here by solving the model, since it is not an equilibrium problem as the ones usually defined when one has a spin-Hamiltonian. Does this mean, that they simulated some kind of dynamics by implying in a non-rigorous manner a transition rule that is given by the Metropolis transition rate?

Reply:

We solved numerically the dynamic equations by mean-field theory and in this paper, in the same spirit we search for the numerical solutions of Eqs.(3)-(5) by MC simulations. This is not an equilibrium statistical mechanics model.

  1. The authors have omitted to discuss many earlier econo-physics approach to such problems. Many other opinion-formation and voter type models were elaborated, some of them allowing also for analytical solution too. One should discuss the results in relation to these results also.

Reply:

Many references to social science and sociophysics articles relevant to our study are included in the first paper (Ref. 45). References cited in the present paper are those closely related to opinion changes.  However, there is no approach similar to our model and our method. So, quantitative comparison is not possible.

REPLY TO REVIEWER 3

Reviewer 3:

In this paper MC simulations are used to explain and quantitatively analyse polarization scenarios that might happen in the community of agents belonging to three different factions. An analogy with the US political situation is exploited for a real-world interpretation, where three factions correspond to Republicans, Democrats, and Independents. The Authors reply on a sociophysics methodology and perform their analysis on a base of a continuous spin Ising model, with spins of three different ‘species’. Each specie spins interact via Hamiltonian (2) whereas inter-specie interactions are governed by Eqs. (3) – (5), here and below I follow the numbering of the manuscript. The model (it was suggested by the present Authors in their preceding publication [44]) contains 12 (!) parameters and its dynamics is afterwards analysed for the case when the spins are located on the sites of a triangular lattice. Here lies the novelty of the results presented in the paper, since formerly the same model was analysed on a complete graph (Kac-like spin model).

I do not have principal objection to the methodology used in the paper. Moreover, the Authors are well known experts in the field of statistical physics, numerical simulations and sociophysics. However, I want to share my opinion left after reading the manuscript. Since the paper is a follow-up of the Ref. [44], the Authors had an obvious challenge met in such situation: on the one hand, one has to repeat some of the statements of an original publication, on the other hand one should not double them by copy/paste. So far, as an interested reader, I miss some relevant info that would allow me better understanding of the results (how 12 parameters are chosen, why some specific values are used for them, why triangular lattice is of specific interest, how the real-world data allows to fit the model to the observations, etc.).
My original intention was to suggest the Authors merging Refs. [44] and the present publications as long as they appear in the same journal issue, but now I found Ref. [44] available online. Therefore, I leave the question open and suggest the Authors to make this paper more self-consistent allowing its reading without checking missing information from Ref. [44]. After such amendments are done I think the paper can be published in Entropy.

Reply:

This paper is not a follow-up of the Ref. 44(now Ref. 45): the model in Ref. 44 (45) is a long-range interaction treated by the mean-field theory while the present paper uses a short-range  interaction model (interaction between NN) and we perform MC simulations to study it. Although the problem of political polarization is the same, the models (long-range vs short-range interaction) and the methods (mean-field vs MC simulation) are different. The agreement between the two methods is remarkable. This will lend support for the use of the mean-field theory in more complicated cases.

For the references, the present papers cite relevant works related to the political polarization. It is self-contained in this aspect.

Reviewer 3 asks how the values of 12 parameters are selected, and why some specific values are used for them; why the triangular lattice is of specific interest; and, how the real-world data allow fitting the model to the observations.

Reply:

With respect to parameter value selection: guided by media and professional, longitudinal polling reports, we assigned parameter values such that they qualitatively mimic general polls results [refs. 43,44].  We have inserted this phrase in the revised version at the first part of Section III. Thanks.

Regarding the choice of lattice, here we have applied the Monte Carlo methodology testing it on a relatively simple geometry: the triangular lattice. The choice of this lattice allows for a maximum number (6) of nearest neighbors (NN) in 2D. Of course, we can use a 3D lattice to have more NN (such as a FCC or a HCP with 12 NN), however we believe it will not give new phenomena. This phrase is added in red characters in the second paragraph of section II.B.

Given our results, this methodology can be applied in the future to complex networks with realistic geometries such as a random network with a variable number of NN. This is stated in the Conclusions section.

Reviewer 3 indicates some errors:
line 151: Democrats’ stance < S1 >= 1 → Democrats’ stance < S1 >= -1
line 233: Si > 2 → Si>^2

These errors have been fixed in the revised version. Thanks.

Reviewer 2 Report

Methods and models of statistical physics proved to be in general useful for studying socio-economic phenomena. The present work intends to apply one of the basic models, the Ising model, for understanding social polarization. The two party polarization is more and more common in the political life of Western societies. The model elaborated previously by the authors, and applied also here intends to describe a system where there are two polarizing extremes and a group of independent voters, who’s political preference might be capable to decide a hypothetical election. Individuals are characterized by a continuous state variable (S) that describes their political preferences in the two-party system. The state-variables are considered as continuous classical spins that interact according to an Ising Hamiltonian, and they are also experiencing an external driving implemented in the model as an external magnetic field. The elements and their S variable are grouped in three categories: the two polarized groups (1) and (2) and the independent group (3). There is a local interaction between the elements in a group arranged on a triangular lattice, and a mean-field like interaction with the other groups. The Hamiltonian of the system is studied in a heat-bath, considering a Metropolis Monte Carlo method. The problem and approach is interesting, however I found many unclear issues, detailed below. Unfortunately In view of these, I cannot recommend publication  of this work. My main concerns are:

1.     The system that is considered here has 12 free parameters (H_i,J_i, K_ij) + Temperature.  The system is studied for one given parameter set, and I found no evidence for a general discussion on the influence of the model parameters, except the Temperature. Fixing the parameters are extremely subjective, and it shows an unexplained asymmetry regarding the interaction with the Independents for the two polarized groups: K_13=0, K_31=-3, K_23=0 and K_31=0. One should explain this asymmetry and also discuss the influence of the model parameters in detail.

2.     The Metropolis Monte Carlo method is designed and used to study equilibrium phenomena where detailed balance is preserved. It generates trustable average values for the relevant physical quantities in equilibrium conditions. The phenomenon studied here is clearly a non-equilibrium problem, where Metropolis dynamics is not justified. One could use the kinetic MC method or the Glauber dynamics, but should not use the Metropolis dynamics.

3.     One of the most problematic issue in this approach is that the elements (and their state variables) are assigned in groups from the beginning and this CAN NOT CHANGE (at least as I understood). For example an element from group (3) cannot leave the group and be reassigned to group (1) or (2). The same for the elements in group (1) or group (2).  Their state variable will/can change, but not the group to which they belong. This means that for instance in group with initially S >0 polarization if an element would change its state variable to S<0, it would still remain in this group and interact with the fixed parameter set according to the Ising Hamiltonian. This is highly unrealistic.

4.     I did not understood (or I missed it) how initially the elements are arranged on the triangular lattice. Elements in the same group are clustered, or independently of their group are randomly positioned? This issue should be clarified. In reality of course one might expect some spatial clustering I for the interaction network based on their political preferences.

5.     In line 159 the authors state” We solved the model….”. It is unclear for me what it is meant here by solving the model, since it is not an equilibrium problem as the ones usually defined when one has a spin-Hamiltonian. Does this mean, that they simulated some kind of dynamics by implying in a non-rigorous manner a transition rule that is given by the Metropolis transition rate?

6.     The authors have omitted to discuss many earlier econo-physics approach to such problems. Many other opinion-formation and voter type models were elaborated, some of them allowing also for analytical solution too. One should discuss the results in relation to these results also.

Author Response

(The authors gave the same response as above.)

Reviewer 3 Report

See the pdf file attached

Author Response

(The authors gave the same response as above.)

Round 2

Reviewer 2 Report

I have studied the new version of the manuscript and the author’s reply letter. My concerns are not answered satisfactorily, therefore I still cannot recommend publication of this work.

I will detail below my main points.

1.     Concerning the non-symmetric K_{ij} inter-group interaction paramaters. I think the authors misinterpreted my concerns. I had and have no problem in breaking the symmetry in the interaction, i.e K_{ij}\ne K_{ji}, which is normal in social interactions. My concerns here was for breaking the symmetry of the interaction of group 3 (Independents) with group 1 (Democrats) and group 2 (Republicans). They propose the following interaction scheme:

               K12 = 4, K21 = 5, K13 = 0, K31 = 3, K23 = 0 and K32 = 0.

This means that they allow for the Independents to interact with Democrats and forbidden the interaction with Republicans. This is a very strong bias in the dynamics, a complete game changer. Definitely such asymmetry will result in a dynamics of Independents towards Democrats, thus a trivial opinion formation. Whys such asymmetry in this interaction towards the two polarizing parties? This was my original concern.

2.     Regarding the Metropolis update algorithm. Detailed balance is important only when one computes equilibrium averages. Both the Glauber and Metropolis update satisfies detailed balance and will drive the system in between the microstates so that the observed averages are OK. For a dynamical process (like the one considered here) one should use properly defined transition rates (as in a kinetic Monte Carlo process), or in case one would like to borough some dynamics from equilibrium MC calculations, the Glauber dynamics is used. The reason is that it is believed for a magnetic system, that the dynamics in such case is realistic. Definitely nobody know what should be use for a social system, but if one would insist on an analogy with the spin system the Glauber update rule is more accepted.

3.     We still do not understand the behavior in the parameter state of this model. I understand that such studies in the mean-field approach were done in the authors previous works. I have checked their work published in 2022 in Entropy and I did not find such study. One should probably understand first the complex parameter space of this model, before studying a very particular choice of the parameters. In such a view we have a common point with Referee 3.

4.     I still do not consider realistic that the groups are static, i.e.  contains the same number of individuals in time, and this cannot change. Once the opinion is changed, it would be realistic to leave this group and be assigned to the one that represents this new opinion.

5.     Seemingly there is a kind of critical temperature for group 3 also: T_3=8.4. Is this induced by the coupling to group 1? Since J_3=0, the only interaction this group experiences is the one with group 1. Maybe, I missed it, but what is Q in Figure 3. One should also define clearly X in Figure 4 and state what is the relation of E (from Figure 2) with the Hamiltonian in equations (3)-(5).

6.     One should not denote both the Hamiltonian (H_i) and the external field (H_i) with exactly the same characters.

Author Response

REPLY TO REVIEWER 2 (second report)

We are thankful to the reviewer for his relevant remarks which allowed us to improve the  text by adding several paragraphs in the new revised version to clarify the points raised by the reviewer.  The new paragraphs are in red characters in the  new version.

Hereafter, we reply to his remarks.

REVIEWER 2, report 2

I have studied the new version of the manuscript and the author’s reply letter. My concerns are not answered satisfactorily, therefore I still cannot recommend publication of this work.

I will detail below my main points.

1.     Concerning the non-symmetric K_{ij} inter-group interaction paramaters. I think the authors misinterpreted my concerns. I had and have no problem in breaking the symmetry in the interaction, i.e K_{ij}\ne K_{ji}, which is normal in social interactions. My concerns here was for breaking the symmetry of the interaction of group 3 (Independents) with group 1 (Democrats) and group 2 (Republicans). They propose the following interaction scheme:

               K12 = −4, K21 = −5, K13 = 0, K31 = −3, K23 = 0 and K32 = 0.

This means that they allow for the Independents to interact with Democrats and forbidden the interaction with Republicans. This is a very strong bias in the dynamics, a complete game changer. Definitely such asymmetry will result in a dynamics of Independents towards Democrats, thus a trivial opinion formation. Whys such asymmetry in this interaction towards the two polarizing parties? This was my original concern.

REPLY:  The choice of parameters is guided by polls  (Refs. 43-44) and the political reality:  Independents (group 3) have no cohesion (J3 = 0) because they have no structure

or means of identifying with each other, they do not communicate, and do not recruit;

therefore, they exert no influence on the other two groups and, as such, K13 = K23 = 0;

 Independents tend to be contrarian to the party in power (here, group 1), thus K31 < 0, and are not influenced by the opposition party, thus K32 = 0. Other examples of contrarian used in a model can be seen for instance in the new references [50-52].

Since the  reviewer did not find our first paper using the mean-field theory, he did not see the above statement and those references. To be self-contained, we rephrase it and include in this new version (red characters at the beginning of Section III) with these new references.

2.     Regarding the Metropolis update algorithm. Detailed balance is important only when one computes equilibrium averages. Both the Glauber and Metropolis update satisfies detailed balance and will drive the system in between the microstates so that the observed averages are OK. For a dynamical process (like the one considered here) one should use properly defined transition rates (as in a kinetic Monte Carlo process), or in case one would like to borough some dynamics from equilibrium MC calculations, the Glauber dynamics is used. The reason is that it is believed for a magnetic system, that the dynamics in such case is realistic. Definitely nobody know what should be use for a social system, but if one would insist on an analogy with the spin system the Glauber update rule is more accepted.

REPLY:  The reviewer said that the Glauber dynamics is more accepted. We agree with his remark. For a comparison between the Metropolis and Glauber dynamics, please see the short note in https://en.wikipedia.org/wiki/Glauber_dynamics. But we would like to draw his attention to the fact that the Glauber dynamics was devised only for Ising spins (the same for Kawasaki dynamics) but not for continuous spins used in our paper. The Metropolis dynamics has the advantage over the Glauber one is that it  does not depend on the nature of spin whether they are Ising, XY, Heisenberg, or continuous Ising spins,…(some recent works tried to generalize the Glauber dynamics to the case of discrete Ising-like spin S (S=-S, -S+1,…, S-1, S) since in this case  one can still calculate  the  Glauber flipping probability, but in the absence of exact solutions for these cases, their validity for general Ising spin S is not yet checked).    So, in our model with continuous spin between -1 and +1, there is no alternative.  We add this remark in the new revised version (in red characters) at the end of Section II.

3.     We still do not understand the behavior in the parameter state of this model. I understand that such studies in the mean-field approach were done in the authors previous works. I have checked their work published in 2022 in Entropy and I did not find such study. One should probably understand first the complex parameter space of this model, before studying a very particular choice of the parameters. In such a view we have a common point with Referee 3.

REPLY: The previous paper using the mean-field theory is fount at the link:

Entropy 2022, 24, 1262. https://doi.org/10.3390/e24091262

We recopy the paragraph how to choose the parameter : 

« To produce anticipatory scenarios of polarization, we made the following assumptions:

_ The Democrats (group 1) are more cohesive than Republicans (group 2), i.e., J1 > J2;

_ Independents (group 3) have no cohesion (J3 = 0) because they have no structure

or means of identifying with each other, do not communicate, and do not recruit;

therefore, they exert no influence on the other two groups and, as such, K13 = K23 = 0;

_ Independents tend to be contrarian to the party in power (here, group 1), thus K31 < 0,

and are not influenced by the opposition party, thus K32 = 0 (see [50–52] for other examples of contrarian used in a model).. »

That choice was based on polls (Refs.43-44) and reflects the reality of political contexts.

We rephrased the above phrase and added to the new version (red characters at the beginning of Section III) together with the new references [50-52] on contrarian models.

4.     I still do not consider realistic that the groups are static, i.e.  contains the same number of individuals in time, and this cannot change. Once the opinion is changed, it would be realistic to leave this group and be assigned to the one that represents this new opinion.

REPLY: The cohesiveness of each group is governed by its intra-group interaction.  Due to this interaction with his neighbors, he stays in the stance of  the party (negative for Democrats, positive for Republicans). His average stance value cannot be otherwise. Even in the dynamical regime, we can still see this in Fig. 5. So the model conserves the population of each party.  Now, for Independents which play the role of contrarians to the governing party (Democrats), their stance has the same sign to that of Republicans (Fig. 5), even if they do not interact with this opposition party.  We can of course construct a model which does not conserve the party populations. But this goes beyond the present work. It may be subject of a future study.  We have added the following phrase in the Conclusion “Note that our present model conserves each party population. It would be interesting to define the conditions under which a member of a given party can leave his party. Though rare, this corresponds to a reality. This will be considered in a near future. »

Thank you for your remark.

5.     Seemingly there is a kind of critical temperature for group 3 also: T_3=8.4. Is this induced by the coupling to group 1? Since J_3=0, the only interaction this group experiences is the one with group 1. Maybe, I missed it, but what is Q in Figure 3. One should also define clearly X in Figure 4 and state what is the relation of E (from Figure 2) with the Hamiltonian in equations (3)-(5).

REPLY: Yes, the critical temperature for group 3 is due to the interaction with group 1.  Thanks a lot for pointing out the omission of the definitions of E and X.  They are now given in Section II in red characters. Q is defined below Eq. (7).

6.     One should not denote both the Hamiltonian (H_i) and the external field (H_i) with exactly the same characters.

REPLY: Thanks a lot. This is corrected in the new version. The external field is now h_i.

Reviewer 3 Report

I do not have further suggestions and recommend the paper to be published as is.

Author Response

The reviewer 3 did not raise any remarks in his second report and recommended the publication of our paper. Thanks.

Round 3

Reviewer 2 Report

Satisfactory improved now, thanks.